# Nitrate Modulates Fruit Lignification by Regulating *CgLAC3* Expression in Pomelo

**DOI:** 10.3390/ijms26094158

**Published:** 2025-04-27

**Authors:** Changhong Lai, Huiwen Zhou, Hong Liao

**Affiliations:** Root Biology Center, Fujian Agriculture and Forestry University, Fuzhou 350002, China; chable518@126.com (C.L.); zhouhw@yeah.net (H.Z.)

**Keywords:** nitrate, lignification, *CgLAC3*, pomelo

## Abstract

Lignification of juice sacs is a primary contributor to reductions in fruit quality, with impacts on taste and economic value of pomelo (*Citrus grandis*). To date, information on the regulation of fruit lignification remains fragmentary. In this study, we first analyzed the relationship between lignification and nutrient status of pomelo juice sacs, which revealed a significant positive correlation between nitrate (NO_3_^−^) concentration and lignin concentration, with over 60% of lignin accumulation explained by NO_3_^−^ levels in three models of machine learning-based regression. Results from field trails in 11 pomelo orchards, as well as in pear fruits and soybean roots exposed to low or high NO_3_^−^ supplies, further demonstrated that nitrate plays an important role in lignification. Transcriptomic analysis further showed that pomelo *laccases* (*CgLACs*) were more intensively up-regulated upon addition of NO_3_^−^ than any of the genes encoding one of the other 12 enzymes involved in lignin biosynthesis. Among the nine identified *CgLACs*, *CgLAC3* was the most significantly up-regulated *CgLAC* in high nitrate treated plants. Over-expressing *CgLAC3* increased lignin concentrations in both pomelo albedo and soybean hairy roots. Taken together, we conclude that nitrate modulates fruit lignification in pomelo through regulation of *CgLAC3* expression, which suggests that NO_3_^−^-N fertilization may affect fruit lignification, and thereby can be managed to improve fruit quality.

## 1. Introduction

Citrus (*Citrus* L.) fruits, with consumer-pleasing juicy textures and sweet flavors, are the most widely distributed fruits in the world, accounting for approximately 40% of global fruit production [1]. In China, citrus fruits comprise one third of total fruit production, making them the veritable king of fruits [1,2]. Pomelo citrus (*Citrus grandis* L.) trees are mostly cultivated in South China, where they comprise 10% of citrus fruit production and are a main source of income for local farmers [1,2,3]. However, in recent years, with increasing cultivation and value of pomelo, farmers have focused on high yields while neglecting declines in fruit quality, which may quickly shrink fruit markets [4,5]. Therefore, quality improvements must be prioritized along with yield in pomelo production. Juice sacs lignification has been reported as a crucial factor for fruit quality [6,7,8]. Typically, it is considered a physiological disorder occurring during late ripening or storage in certain fruits, such as pear (*Pyrus* spp.) [9,10,11], loquat (*Eriobotrya japonica*) [12,13,14,15], and citrus [8,16,17,18]. Once fruits lignify, significant declines in quality manifest as hard peels, water depleted flesh, and reduced sweetness and acidity [18]. Excessive lignin deposition within juice sacs has been intimately linked with physiological disorders in fruits [7,8,18]. Thus, modulating fruit lignification might be a means to maintain its quality. In addition, the findings shed light on the physiological and biochemical responses of tobacco plants to NO_3_^−^ treatment, with it being observed that the cell wall development of leaves from low NO_3_^−^ supply was incomplete, and the cell wall from high NO_3_^−^ supply concentration was notably thicker [19]. However, the response of NO_3_^−^ supply to lignification in pomelo fruit is still unclear.

Many studies have demonstrated that lignin is synthesized through phenylpropaniod metabolites and their 12 related enzymes, including phenylalanine ammonia-lyase (PAL), cinnamate 4-hydroxylase (C4H), 4-coumarate-CoA ligase (4CL), *p*-coumarate 3-hydroxylase (C3H), shikimate/quinate hydroxy cinnamoyl transferase (HCT), caffeoyl-CoA 3-O-methyltransferase (CCoAOMT), caffeoyl shikimate esterase (CSE), ferulate-5-hydroxylase (F5H), cinnamoyl-CoA reductase (CCR), cinnamyl alcohol dehydrogenase (CAD), polyamine oxidase (PAO), peroxidase (PER), and laccase (LAC) [20,21,22]. Among them, LACs are responsible for the critical last step of polymerizing lignin monomers constructed from *p*-coumaryl alcohol, coniferyl alcohol, and sinapyl alcohol into lignin polymers [22,23,24,25]. Wide distribution of lignin among plant tissues, and essential roles for *laccase*s in physiological and biochemical processes draw the attention of researchers towards the roles of this biosynthetic pathway and its final polymerization step. For example, simultaneous disruption of *AtLAC4*, *AtLAC11*, and *AtLAC17* reduced lignification and impaired both vascular and whole plant development in *Arabidopsis thaliana* [24]. The extracellular domain of another *Arabidopsis laccase*, *AtLAC3*, governed casparian strip development [26]. Meanwhile, in cotton production, *GhLAC1* modulated fiber initiation and elongation by coordinating jasmonic acid and flavonoid metabolism [27]. While other *LAC*s are more widely expressed, they may play specific roles in lignin polymerization [20].

Fruit production may also involve lignin biosynthesis and polymerization. For example, during apple fruit development, the expression of *LAC*s significantly increases and aligns with lignin deposition [28,29,30]. In pear fruits, silencing *PbrLAC1* reduces lignin content [31]. However, the regulation and specific molecular participants involved in fruit lignification remain largely unclear. Interestingly, nutrient management affects lignification. For example, applying silicon (Si) fertilizers to rice enhances the expression of *OsCAD*, and thereby increases lignin accumulation, stalk strength, and lodging resistance [32]. Conversely, excessive nitrogen (N) fertilization significantly reduces the mechanical strength of stalks and lodging resistance in buckwheat and japonica rice by inhibiting lignin biosynthesis [33,34,35]. However, Kong et al. find that potassium (K) addition effectively mitigates the adverse effect of excessive N fertilization on wheat culm strength and lodging resistance through increasing lignin accumulation in vascular bundles [36]. In addition, N deficiency has been reported to reduce root solidity through decreasing rapeseeds root lignin concentrations [37]. These reports suggest that the involvement of nutrients in lignification is a common phenomenon among plant species and tissues, not just fruits. However, molecular mechanisms underlying lignification have yet to be elucidated for many plant species and production systems. In this study, we first evaluated the contribution of NO_3_^−^-N to lignin accumulation in pomelo fruits. Then, we performed several physiological and molecular experiments to decipher the regulatory mechanisms modulating observed NO_3_^−^-N effects on lignification.

## 2. Results

### 2.1. Nitrate Is an Important Determinant of Fruit Lignification

Previously, within each pomelo fruit, we have observed the structure of pomelo fruits including flavedo (F), albedo (A), segment membrane (S), and juice sacs which had both vertical juice sac (Jv) and transverse juice sac (Jt) orientations [4] (Appendix A). Here, we determined that the nutrients included total nitrogen (TN), total phosphorus (TP), total potassium (K), soluble nitrogen (NO_3_^−^-N, NH_4_^+^-N and amino acid nitrogen (AA-N)), and soluble phosphorus (SP). The results showed that NO_3_^−^-N concentration in the Jv exhibited statistically significant differences under continuous two-year nutrient management regimes (Figure 1 and Appendix A), suggesting that it plays an important role in pomelo quality. We further determined following previous lignification that the Jvs were more readily lignified than Jts, as indicated by much higher lignin concentrations in the Jvs than in Jts (Figure 2B). One implication is that Jvs are more prone to lignification due to excessive accumulation of lignin than Jts. With these observations in hand, we focused thereafter on studying Jv lignification in pomelo fruits.

Correlation analysis was carried out to elucidate any relationship between Jv lignification and nutrient concentration in pomelo fruits. Here, lignin concentration was significantly positively correlated to NO_3_^−^-N, NH_4_^+^-N, and soluble P (SP) concentrations, but negatively correlated to total N (TN), total P (TP), and amino acid N (AA-N) (Figure 2C). Furthermore, three models of machine learning-based regression were separately employed to deduce the relative importance of different nutrients on lignin concentration. The results showed that NO_3_^−^-N, SP, TP, and TN were the top four factors affecting Jv lignin deposition, with NO_3_^−^-N ranking first based on an importance ratio for Jv lignin accumulation of 61.2–69.9% (Figure 2D). This suggests that NO_3_^−^-N might be an important nutrient for determining Jv lignification in pomelo fruits.

Field trails with low N (LN) or high N (HN) treatments were performed in 11 pomelo orchards to evaluate the relationship between lignification and NO_3_^−^-N. The Jv concentrations of lignin and NO_3_^−^-N under HN conditions were 86.13% and 48.84% higher, respectively, than in LN treatments (Figure 3A,B). Furthermore, lignin concentration was significantly positively correlated to NO_3_^−^-N concentration (Figure 3C). These results confirmed that nitrate was a pivotal nutrient for determining the extent of lignification in the Jv of pomelo fruits.

To test whether NO_3_^−^-N is associated with lignification in other fruits, as well as in other non-fruit plant tissues, we exposed pear pulp and soybean roots to LN and HN treatments. Both pear pulp and soybean roots became more lignified in HN, as indicated by deeper shades of red through in situ staining with Wiesner’s reagent, along with 45.82% and 43.48% higher lignin concentrations in HN than in LN for pear pulp and soybean roots, respectively (Appendix A). Taken together, the results outlined above demonstrate that high NO_3_^−^-N promotion of lignification is common in plants.

### 2.2. Candidate Genes Involved in Regulating Nitrate Effects on Lignification

To evaluate the molecular machinery underlying nitrate effects on lignification, transcriptome analysis was conducted with the Jv of pomelo fruits exhibiting varying degrees of lignification under LN and HN treatments in field trails. Consistently, the Jvs of pomelo fruits from HN treated plants contained much higher lignin and NO_3_^−^-N concentrations than those from LN-treated plants (Figure 4A,B). Expression of genes encoding 12 enzymes involved in the lignin biosynthesis was further analyzed through RNA-seq analysis (Appendix A). Among them, expression of *laccase* (*LAC*) genes responded the most to different NO_3_^−^-N treatments (Figure 4C). We also quantified the activity of LACs in the Jv of pomelo fruits, and found that LAC activity was significantly higher in HN than that in LN treatments (Figure 4D). These results suggest that LACs and their encoding genes might be the key modulators of NO_3_^−^-N impacts on fruit lignification in pomelo.

Based on genome sequence, a total of 9 *LAC*s were identified in the pomelo genome, and the RNA-seq data showed that all nine of the *CgLAC*s were up-regulated by HN addition, which was also confirmed by quantitative real time PCR (qRT-PCR) measurements (Figure 4E). Among pomelo *laccase* genes, the expression of *CgLAC3* was most significantly responsive to NO_3_^−^-N status, as shown by a 5.45-fold greater expression in HN than in LN, which implies that *CgLAC3* could be a key gene in NO_3_^−^-N regulation of lignification in pomelo fruits.

### 2.3. Over-Expression of CgLAC3 Promotes Lignification in Both Pomelo Albedo and Soybean Hairy Roots

Phylogenetic analysis revealed that *CgLAC3* is closely linked to *AtLAC3* (Figure 5A), suggesting that the two genes might have similar functions in lignin biosynthesis. We constructed a *CgLAC3* promoter (-1941 bp) vector fused with GUS-encoding gene and directly injected this construct into *Nicotiana benthamiana* using a *35S*::GUS vector as the positive control. After staining, injected areas of leaves supplied with HN exhibited distinctively deeper blue shading than LN-treated leaves (Figure 5B). RT-qPCR analysis also revealed that *GUS* expression of *pro CgLAC3::GUS* was significantly higher in HN than in LN (Figure 5C), indicating that the *CgLAC3* is highly induced by high NO_3_^−^ conditions.

To further test whether *CgLAC3* regulates lignification in pomelo fruits, we constructed a *CgLAC3* overexpression (OE) vector fused with a GFP-encoding gene and directly injected this construct into young pomelo peels (Figure 6A). First, GFP signals were detected in the injected section to ensure successful *CgLAC3* transformation (Figure 6B). Then, GFP signaling regions were sampled to determine CgLAC activity and lignin concentration. Consistently, CgLAC enzymatic activity and lignin concentration were significantly higher in OE peels than in empty vector (EV) control peels by 31.34% and 55.86%, respectively (Figure 6C,E). This indicates that *CgLAC3* expression led to CgLAC activity and, indeed, thereby promotes lignification in pomelo peels. This was also confirmed by darker red Wiesner staining in the albedo of OE peels than in EV peels (Figure 6D). The results together with other evidence reported above strongly suggest that *CgLAC3* positively regulates lignin accumulation in pomelo fruits through impacts on CgLAC activity.

To further test the effects of *CgLAC3* expression on lignification, the soybean transgenic hairy roots were also employed for functional analysis. Both *35S::GFP* (EV) and *35S::CgLAC3-GFP* (OE) vectors were constructed and transformed into *Agrobacterium rhizogenes* prior to infecting soybeans with these strains to induce transgenic hairy root formation. After verifying GFP signals in transgenic hairy roots (Figure 6F), in situ staining of lignin and measurements of lignin concentration were carried out. Consistently, overexpression of *CgLAC3* promoted lignification in soybean hairy roots, as evident by staining and measurements of higher lignin concentrations in OE hairy roots than in EV lines (Figure 6G,H), which further demonstrated that *CgLAC3* positively regulates lignin accumulation in planta.

## 3. Discussion

### 3.1. Nitrate Plays an Important Role in Regulation of Lignin Biosynthesis in Plants

Lignin is extremely important for plant growth and development. Its biosynthesis and accumulation are also affected by nutrient management, particularly nitrogen (N) application. However, specifics of how N affects lignification remain contentious in available reports. For instance, excessive N fertilizer application may reduce the mechanical strength of stalks in buckwheat and rice by limiting cellulose and lignin concentrations [38,39], while increasing lignin accumulation in rapeseed roots [37]. Under low N conditions, Wang et al. [38] observed notable declines in apple rootstock lignin concentrations. Many forms of N exist in plants and soils, including organic and inorganic forms [40]. It also was reported that the cell wall from high NO_3_^−^ supply concentration is notably thicker, and the lignin content was lower under lower NO_3_^−^ supply [19]. This variety of N forms plays different roles in plant growth and development [39,40]. Apparent controversy originating from conclusions drawn in studies of N effects on lignification as described above might be clarified by accounting for the specific contributions to plant lignification provided by each of the different forms of N that plants may acquire and utilize.

In this study, we analyzed the concentrations of four forms of N in pomelo juice sacs, including NO_3_^−^-N, NH_4_^−^-N, AA-N, and total N. Here, an important role for NO_3_^−^-N in regulation of lignin biosynthesis in pomelo juice sacs was revealed through estimation of an importance ratio of over 60% for NO_3_^−^-N contributions to lignin accumulation in machine learning-based regression models. Consistently, we observed a significant positive correlation between NO_3_^−^-N and lignin concentration in pomelo juice sacs samples from 11 orchards, as well as increases in lignin accumulation in pear pulp and soybean roots under HN conditions, which further supports the conclusion that NO_3_^−^-N may play pivotal roles in regulation of lignification in plants. As the major inorganic form of N in plants, NO_3_^−^-N has been known to play many critical roles in plant metabolism, growth regulation, signal transduction, and so on [38,41,42]. Our findings further revealed that excess NO_3_^−^-N might cause severe lignification and thus reduce fruit quality. Therefore, we suggest to avoid applying excessive amounts of N in practice so as to reduce over-accumulation of NO_3_^−^-N in order to minimize lignification and thus improve fruit quality.

### 3.2. CgLAC3 Is the Key Nitrate-Responsive Lignification Gene in Pomelo Fruits

Lignification, which leads to granulation in pomelo juice sacs [4,8,43], is involved in many complex biosynthetic processes in plants. When citrus fruits undergo lignification, it can affect hardness, storage durability, and taste, thereby critically limiting fruit quality and shelf life [8,44]. Biosynthesis of lignin in plants has been intensively reported [45]. In total, 12 enzymes, including polymerizing laccases, are known to be critically involved in different steps of lignin biosynthesis [19,20,21,46]. However, information on related regulatory pathways remains fragmentary. Several *LACs* have been directly linked with lignin biosynthesis. For instance, *AtLAC1* and *PbLAC1* are reportedly involved in cell wall lignin deposition [31]. In addition to lignification, *LAC*s can also affect fruit color, as *DkLAC1* polymerizes proanthocyanidins (PAs) in fruit flesh [47], and *PbLAC4* participates in fruit anthocyanin degradation in pears [29]. Different functions are fulfilled by specific *LACs*. *Arabidopsis thaliana*, for one, codes 17 *LAC*s exhibiting low amino acid sequence homology, though copper-binding domains are highly conserved [46,48]. Among the 17 *AtLAC*s, *AtLAC3* specifically expresses in root endodermal cells, where it collaborates with the casparian strip membrane-localized protein (CASP) to provide positional information for the casparian strip, a structure with significant lignin deposition [26]. In *Arabidopsis thaliana*, miR408 is found to target 52 transcripts that encode two distinct categories of copper-containing proteins. These encompass the small blue copper protein known as plantacyanin (PCY), as well as LAC3, LAC12, and LAC13 [49,50,51]. However, the transcript level of PCY, but not that of LAC3, LAC12, and LAC13, was induced by ABA in the seedlings [52]. This can provide a reference for whether ABA can induce the transcriptional level of LAC3 in pomelo for the next step of experimental research. Meanwhile, it was found that *AcLAC35* in kiwifruit leaves resulted in greater lignin content than in wild-type leaves, leading to the formation of thicker cell walls, which could enhance resistance against Psa infection [53]. In this study, we demonstrated that *CgLAC*s play a critical role in nitrate-responsive lignin biosynthesis in pomelo fruits by first revealing that among 13 lignin biosynthesis pathway genes, *CgLAC* expression is the most significantly up-regulated by HN, and then by observing increased LAC activity with increasing NO_3_^−^ availability. Further functional investigation using pomelo fruits and soybean roots identified *CgLAC3* as the candidate LAC gene responsible for NO_3_^−^-N regulation of fruit lignification. Similarly to functions of its homolog *AtLAC3* in *Arabidopsis* [48], overexpressing *CgLAC3* increased lignin accumulation in both pomelo fruits and soybean hairy roots (Figure 6). Finally, it is worth reiterating that *CgLAC3* expression was dramatically enhanced in HN- or LN-treated fruits, further supporting the conclusion that *CgLAC3* is the critical gene responsible for NO_3_^−^-N regulation of lignification in pomelo fruits (Figure 7). However, sorting through details of how nitrate regulates *CgLAC3* and thereby lignification still requires further evaluation.

## 4. Materials and Methods

### 4.1. Plant Materials

Pomelo fruits (*Citrus grandis* cv Sanhongyou) were collected from Wuxing village, Pinghe county, Fujian province, China (117.244737° E, 24.301727° N) in September 2022. Fruits of similar size were randomly collected from 12 trees in one orchard. Four fruits were sampled from each tree, for a total of 48 fruits included in relationship analysis between lignin and nutrients following the methods outlined by Lai et al. [3]. In 2023, field trails were carried out in 11 pomelo orchards with 2 N treatments. Low N (LN) or high N (HN) treatments were applied as 160 N. ha^−1^ or 320 kg N. ha^−1^ in one year, respectively. Six fruits were sampled from each N treatment in each orchard, and a total of 132 fruits were used for lignin and nutrient analysis.

Pear (*Pyrus* spp. cv. Dangshansuli) fruits are rich in stone cells, serving as an excellent model for studying lignification, and were obtained from a supermarket and sampled by Meng et al. [54]. Fruits with similar size were treated with 0.01 (LN) and 0.15 g NO_3_^−^ kg^−1^ (HN), and then were kept at 25 °C for 10 h (Appendix A). Pears were sampled for lignin and nitrate concentration measurements. Uniform soybean (*Glycine max.* cv. Huachun NO.3) seedlings were cultivated in the nutrient solution containing 0.20 (LN) or 20.0 g NO_3_^−^ kg^−1^ (HN) for 16 days. Six soybean roots were sampled for lignin and nutrient concentration determinations.

Tobacco (*Nicotiana benthamiana*) plants serve as an ideal material for transient transformation to study gene expression [55]. They were cultivated in LN (0.10 g NO_3_^−^ kg^−1^) and HN (10.00 g NO_3_^−^ kg^−1^) solution for 20 days prior to randomly selecting leaves to be transformed by *Agrobacterium rhizogenes* with *35S::GUS* or promoter-*CgLAC3-GUS* vectors. Three days after transformation, leaves were stained with GUS staining solution [56], and GUS expression driven by the *CgLAC3* or *35S* promoter was quantified for HN and LN treatments.

Peels of young pomelo (*Citrus grandis* cv. Sanhongyou) were transformed with *Agrobacterium rhizogenes* harboring *35S::GFP* (EV) or *35S::CgLAC3-GFP* (OE) vectors. Three days after transformation, the albedo of young pomelo was sampled for determining lignin concentration and CgLAC activity.

Soybean roots were transformed by *Agrobacterium rhizogenes* to induce hairy roots with *35S::GFP* (EV) or *35S::CgLAC3-GFP* (OE) vectors according to Li et al. [56]. After transgenic hairy roots emerged, they were transferred into nutrient solution containing 0.20 g NO_3_^−^ kg^−1^ (LN) for 16 days before determining lignin and nutrient concentrations.

### 4.2. Determination of Fruit Indicators

#### 4.2.1. Nutrient Determination

For nutrient determinations, total nitrogen (TN), total phosphorus (TP), and total potassium (K) of dried pomelo juice sacs were measured using the H_2_SO_4_-H_2_O_2_ digestion method [3]. TN and TP were measured using a flow analyzer (Skalar San++, Amsterdam, The Netherlands), and K was determined by atomic absorption spectroscopy (AAS). Soluble nitrogen (NO_3_^−^-N, NH_4_^+^-N, and amino acid nitrogen (AA-N)) and soluble phosphorus (SP) of fresh pomelo juice sacs were determined following methods described by Lai et al. [3]. Briefly, NO_3_^−^-N, NH_4_^+^-N, AA-N, and SP were monitored by the salicylic acid colorimetric method, indophenol blue colorimetric method, ninhydrin colorimetric method, and molybdenum antimony ascorbic acid colorimetric method, respectively.

#### 4.2.2. Lignin Staining and Lignin Determination

In situ lignin staining of fresh plant samples, including pomelo juice sacs, pomelo albedo, pear fruit, and soybean roots, took place with Wiesner reagent as Lai et al. [4] and Shi et al. [8] reported. Lignin concentrations of fresh samples were determined following the thioglycolic acid method. After extraction, lignin was dissolved in 1 mL 1M NaOH, and measured at a wavelength of 280 nm, with 1M NaOH serving as the control.

#### 4.2.3. LAC Enzyme Activity Determination

Plant LAC ELISA KIT (Shanghai Yuanju biotechnology center, Shanghai, China) was used to determine LAC enzyme activity as follows: About 1.0 g vertical juice sacs was sampled and ground with liquid nitrogen. Then, extraction medium was added to 1 mL and centrifuged at 4 °C 10,000 r min^−1^ for 15 min. The supernatant of the enzyme liquid was stored at 4 °C before being measured. A total of 10 µL of sample was dispensed into each designated sample wells, followed by the addition of 40 µL of diluent. Then, 100 µL of horseradish peroxidase (HRP)-labeled detection antibody was introduced. The wells were then sealed with a plate sealer and incubated in a thermostat at 37 °C for 60 min. After that, 50 µL of substrate A and 50 µL of substrate B were added to each well, followed by a 15 min incubation at 37 °C in the dark. To terminate the reaction, 50 µL of stop solution was added. Within 15 min, the optical density was measured at 450 nm using a microplate reader of Varioskan™ LUX (Thermo Scientific™, Waltham, MA, USA) operated with appropriate controls.

### 4.3. RNA Isolation and qRT-PCR Analysis

Total RNA was isolated using TRIzol regent according to the RNeasy Mini Kit (TRAN). One microgram of total RNA was used for first-strand cDNA synthesis using a ReverTra Ace RT-qPCR Master Mix kit (Toyobo, Osaka, Japan) following the manufacturer’s instructions.

Gene expression was determined by quantitative real-time (qRT) PCR using Thunderbird SYBR qPCR Mix (Toyobo, Osaka, Japan) on a Mastercycler ep realplex (Eppendorf, Hamburg, Germany), with three biological replicates included for each gene. *EF1α* was chosen as the internal control for all analyses. Gene expression levels were normalized by the 2^−ΔΔCT^ method. Primers in this article are listed in Appendix A.

### 4.4. Data Analysiss

All data were analyzed using IBM SPSS Statistics (version 20.0) Student’s *t*-test and covariance analysis with multi-factor analyses. Methods of XGBoost regression/Decision tree regression/GBDT regression were as follows (https://tianchi.aliyun.com/course/278?spm=5176.21206777.J_3641663050.11.5b7617c9LEQth0Python, accessed on 22 April 2025): (1) Build the regression model through the training set data. (2) Calculate feature importance by establishing its regression. (3) Apply the established regression model to the training and test data to obtain the model evaluation results. Graphical representation was performed using GraphPad Prism 8, R language (R 4.2.2) with the ggplot2 package, and online tools such as Hiplot (https://hiplot.com.cn/cloud-tool/drawing-tool/link/653, accessed on 22 April 2025) and Chiplot (https://www.chiplot.online/, accessed on 22 April 2025).

## 5. Conclusions

Taken together, we conclude that NO_3_^−^-N plays an important role in inducing the lignification of pomelo fruits through analysis of RNA-seq and CgLAC enzyme activity, that a pivotal candidate gene, *CgLAC3*, involved in NO_3_^−^-N regulating lignification was identified, and that the functions of *CgLAC3* in responding to NO_3_^−^-N levels and regulating lignin synthesis were evaluated through transient transgenic transformation technology. Expression of the pro *CgLAC3* in tobacco leaves was able to respond to external NO_3_^−^-N levels, and overexpression of *CgLAC3* significantly increased lignin concentration in the pomelo flavedo and hairy roots of soybean plants. The results here not only uncover NO_3_^−^-N regulation of fruit lignification, but also provide a theoretical basis for managing NO_3_^−^-N fertilization to modulate lignin concentrations and thereby improve fruit quality.

## Figures and Tables

**Figure 1 ijms-26-04158-f001:**
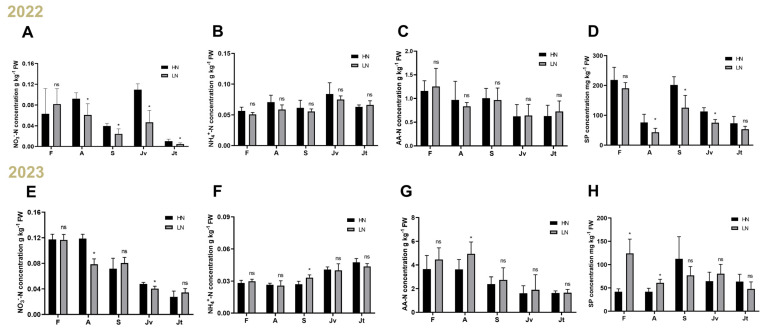
Effects of nutrient management on SN and SP allocation in different parts of pomelo fruits. (**A**) NO_3_^−^-N concentration in 2022; (**B**) NH_4_^+^-N concentration in 2022; (**C**) AA-N concentration in 2022; (**D**) SP concentration in 2022; (**E**) NO_3_^−^-N concentration in 2022; (**F**) NH_4_^+^-N concentration in 2022; (**G**) AA-N concentration in 2022; (**H**) SP concentration in 2022. SN: NO_3_^−^-N, NH_4_^+^-N, AA-N; LN: application with fertilization as 160 kg N. ha^−1^; HN: application with fertilization as 320 kg N. ha^−1^; (**A**–**H**): *n* = 6, Student’s *t*-test. *: *p* < 0.05, ns: no significant difference.

**Figure 2 ijms-26-04158-f002:**
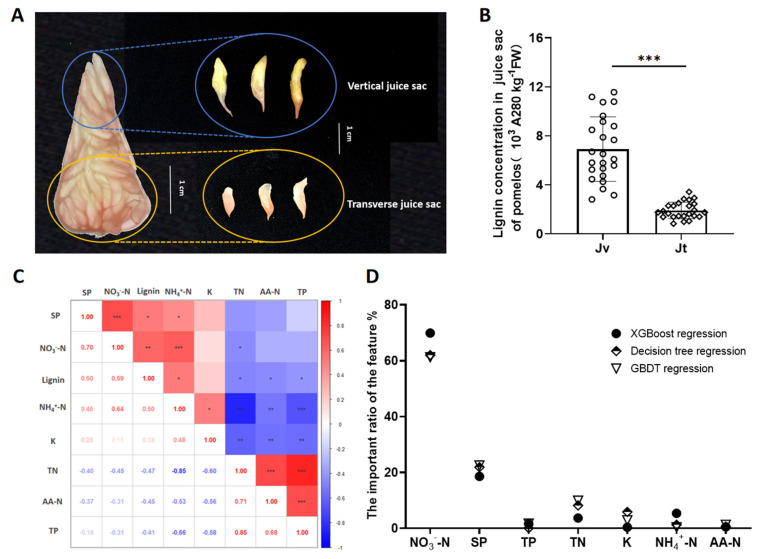
Relationship between lignin and nutrients in pomelo juice sacs. (**A**) Sampling different types of juice sacs; (**B**) lignin concentration; (**C**) correlation analysis among lignin and nutrient concentrations; (**D**) relative contribution of nutrients to lignin accumulation through 3 models of machine learning-based regression deduction. Jv: vertical juice sac; Jt: transverse juice sac; *n* = 48. (**C**): *: 0.01 < *p* ≤ 0.05; **: 0.001 < *p* ≤ 0.01; ***: *p* ≤ 0.001.

**Figure 3 ijms-26-04158-f003:**
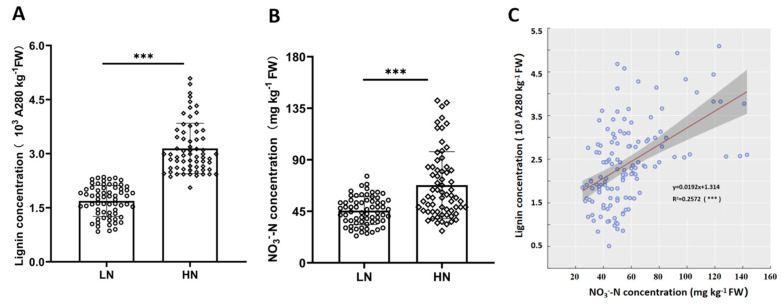
High nitrate supplies increase pomelo juice sac lignin concentrations across 11 orchards in field trails. (**A**) Lignin concentration; (**B**) NO_3_^−^-N concentration; (**C**) correlation between lignin and NO_3_^−^-N concentration. LN: application with fertilization as 160 kg N. ha^−1^; HN: application with fertilization as 320 kg N. ha^−1^. *n* = 132. ***: *p* < 0.001.

**Figure 4 ijms-26-04158-f004:**
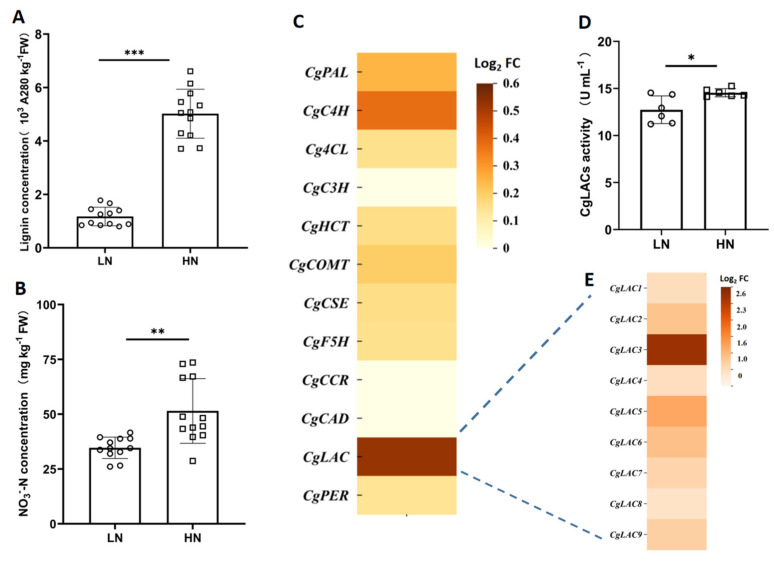
Screening candidate genes for involvement in NO_3_^−^-N regulation of lignification. (**A**) Lignin concentration; (**B**) NO_3_^−^-N concentration; (**C**) heat-map of gene expression differences between HN and LN quantified in RNA-Seq analysis; (**D**) enzyme activity of CgLAC. (**E**) heat-map of CgLAC expression differences observed in RT-qPCR analysis; (**A**,**B**) *n* = 12, (**D**) *n* = 6; *: 0.01 < *p* ≤ 0.05; **: 0.001 < *p* ≤ 0.01; ***: *p* < 0.001.

**Figure 5 ijms-26-04158-f005:**
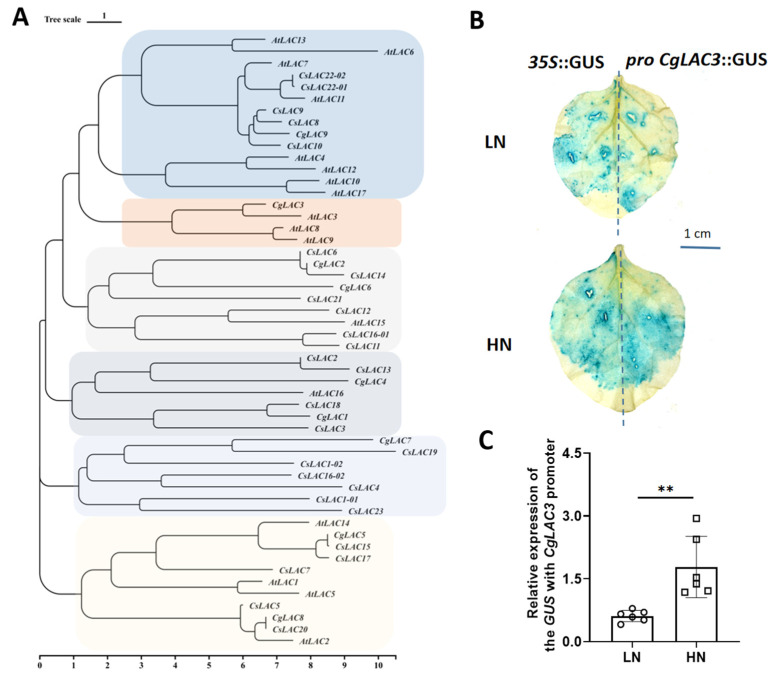
Phylogenetic analysis of *CgLAC3* in plants and its promoter responses to nitrate. (**A**) Phylogenetic analysis of *laccase* genes from pomelo and *Arabidopsis*; (**B**) GUS staining of transgenic tobacco leaves; (**C**) GUS expression with *CgLAC3* promoter. *At*, *Arabidopsis thaliana*; *Cs*, *Citrus sinensis*. LN: 0.10 g NO_3_^−^ kg^−1^; HN: 10.0 g NO_3_^−^ kg^−1^. (**C**) *n* = 6. **: *p* < 0.01.

**Figure 6 ijms-26-04158-f006:**
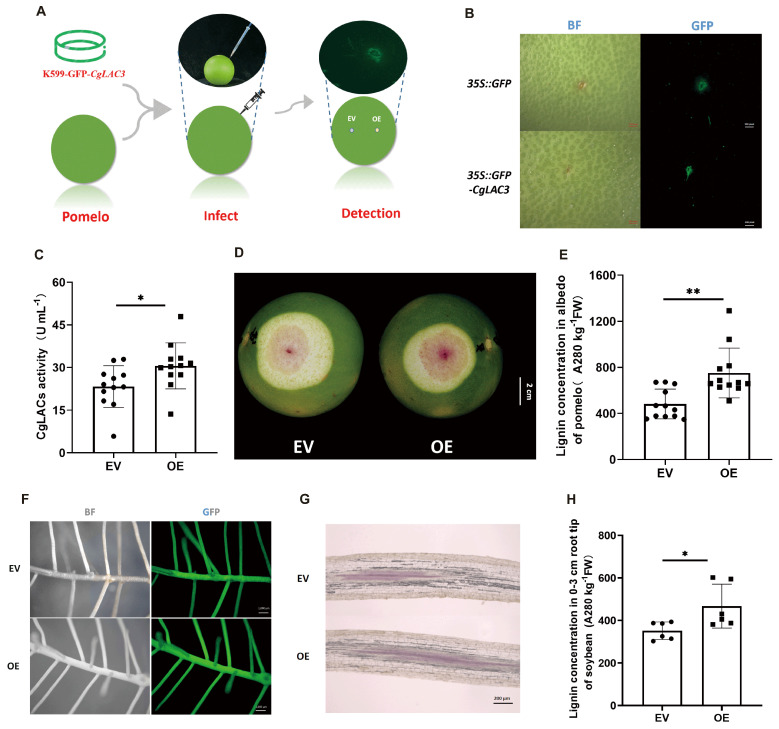
Overexpressing *CgLAC3* promotes lignification in both pomelo peel albedo and soybean hairy roots. (**A**) Schematic diagram of *CgLAC3* transformation in young pomelo peels; fluorescence signal detection in pomelo peels (**B**) and soybean hairy roots (**F**); (**C**) CgLAC enzyme activity; in situ Wiesner staining of lignin in pomelo albedo (**D**) and soybean hairy roots (**G**); Lignin concentration in pomelo albedo (**E**) and soybean hairy roots (**H**). (**C**,**E**) *n* = 12, (**H**) *n* = 6; *: 0.01 < *p* ≤ 0.05 and **: 0.001 < *p* ≤ 0.01.

**Figure 7 ijms-26-04158-f007:**
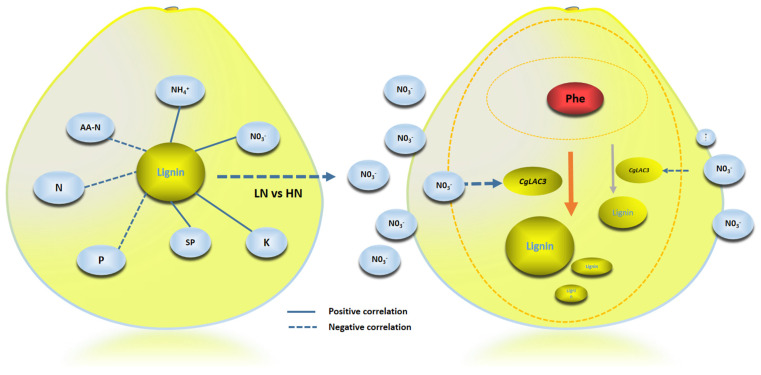
A proposed model illustrating the function of NO_3_^−^-CgLAC3-Lignin.

## Data Availability

The original contributions presented in this study are included in the article/Appendix A; further inquiries can be directed to the corresponding author.

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
