# Peer review of "Nitrate Modulates Fruit Lignification by Regulating CgLAC3 Expression in Pomelo"

_ijms, 2025, doi:10.3390/ijms26094158_

Round 1

Reviewer 1 Report

Comments and Suggestions for Authors

Dear Authors,

Please find my recommendations in the next:

  1. The title didn’t properly reflect what is described in the manuscript (the authors in the methods section describe experiments performed on pear, tobacco, soybean, etc.)
  2. The introduction should better describe the current status and gaps in knowledge related to the field of fruit lignification:
  • L37-43: The manuscript didn’t clearly present the tissue-specific regulatory mechanisms and NO3- accumulation pattern in citrus species fruits
  • L44-65: The manuscript should present details on post-translational regulation of laccases
  • L66-67: The manuscript should more clearly present the relationship between NO3- concentration and lignification thresholds; also, the manuscript should consider, in my opinion, the temporal dynamics of NO3- induced responses
  • L76-82: Considering those presented in the introduction and the hypothesis/objectives of presented research manuscript, in my opinion, there are weak connections established between NO3- signaling and laccase activation; also, the manuscript should better justify the selection of CgLAC3
  1. L257: In the methods section, the manuscript presents the studied plant materials: Pomelo fruits (Citrus grandis cv Sanhongyou) – L 258; Pear (Pyrus spp. cv. Dangshansuli) – L268; Uniform soybean (Glycine max. cv. Huachun NO.3) – L270; Tobacco (Nicotiana benthamiana) – L274; In my opinion, in the manuscript should justify and motivate:
  • Justification: Why the authors has selected these specific heterologous systems
  • Motivation: The significant phylogenetic distance between pomelo and the other species didn’t compromise direct functional comparisons
  • How the different ways of tissue organisation and developmental patterns affect lignification processes
  1. L267-270: It is not clear how pears were treated with NO3- (immersions, spraying?). The authors should present clear details for:
  • Please justify the selection of these concentration (0.01 and 0.15 g NO3-; L268)
  • Why soybean was exposed to other concentration (0.2 and 20  g NO3-; L 271)
  • Details on growth conditions are missing (light intensity, photoperiod, humidity, temperature, etc)
  • What standardization metrics were considered for plant developmental stages?
  • When case, please provide details about soil composition and sterilization procedures
  • For laboratory experiments what acclimation period was considered before treatments? Also please detail how was verified (if it was verified) the NO3- uptake rate; and if pH adjustment was considered or not for the nutrient solution;
  1. L300: In my opinion, different tissue types used across species (roots vs. fruits) complicate comparative analyses; the authors should present how they overcome these challenges both in applied analytical methods and in results relevance and interpretation/comparisons. Considering the applied method, what kind of quality control assurance was performed considering the varied tissues/species?
  2. L296-299: The authors should be more specific; they should also provide details about control and the number of replicates considered
  3. L 83: Considering the “Results section” in my opinion:
  • Control groups are not well presented and compared. This diminishes the assessment of treatment-specific effects relevance;
  • Time-dependent changes are nnot clearly presented;
  • Overall, the results lack detailed comparisons between different tissues or conditions that could provide further mechanistic insights
  1. For the discussion section (L198) in my opinion, the authors should:
  • more adequately and accurately integrate and compare their results with recent literature;
  • more better consider alternative biochemical pathways that influence lignification;
  • proposed mechanistic insights, specifically regarding NO₃⁻ and CgLAC3, shoul provide more detailed molecular explanations;
  • the extrapolation from gene expression to functional outcomes should more adequately justified and provide supporting evidence
  • better consider the time-dependent aspects of lignification responses, in my opinion this is crucial for dynamic process interpretation
  1. L340-344: The conclusion section should be more clear and accurately presented; in my opinion, in the current version of the manuscript, this is poorly treated by the authors

Author Response

Dear Editors and Reviewers:

Thank you very much for the valuable comments on our manuscript entitled “Nitrate modulates fruit lignification by regulating CgLAC3 expression in pomelo” (Manuscript ID: 3595479). Those comments are all valuable and very helpful for revising and improving our paper, as well as the important guiding significance to our researches. We have studied comments carefully and have made revisions under revision mode in the resubmitted manuscript which we hope meet with approval. The corrections in answer: to the reviewer’s comments are listed one by one as follows:

Reviewer 1 

Comments1: The title didn’t properly reflect what is described in the manuscript (the authors in the methods section describe experiments performed on pear, tobacco, soybean, etc.)

Response1: Thank you for your question and suggestion. In order to more clearly express the mechanism study of the lignification of pomelo fruits, we changed Figure 1 as the nutrient concentration distribution, with pear and soybean as supplementary materials. Therefore, main results are most from pomelo and the title should be OK.

Comments2: The introduction should better describe the current status and gaps in knowledge related to the field of fruit lignification:

Response2: Thanks for your advice. We considered the following aspects when composing the introduction section: 1. The distribution and production of Citrus fruits; 2. Lignification is an important factor affecting fruit quality and which fruits lignification has been reported; 3. The pathways of lignification; 4. The influence of nutrients on lignification.

Comments3: L37-43: The manuscript didnt clearly present the tissue-specific regulatory mechanisms and NO3- accumulation pattern in citrus species fruits

Response3: Thank you for your question. This paragraph is to explain what fruit is prone to lignification, which is an important factor affecting quality (L37-43) and  we added the finding shed light on the physiological and biochemical responses of tobacco plants to NO3- treatment in the line 44-48. We also present nutrient regulatory mechanisms in the line 72-76. 

Comments4: L44-65: The manuscript should present details on post-translational regulation of laccases

Response4: Thanks for your questions and advice. We present LACs function which have been reported in the line 60-65.

Comments5: L66-67: The manuscript should more clearly present the relationship between NO3- concentration and lignification thresholds; also, the manuscript should consider, in my opinion, the temporal dynamics of NO3- induced responses

Response5: Thanks. The temporal dynamics of NO3- induced responses is worthy to study but here we more focus on whether the CgLAC3 promoter region responds to NO3- in the Figure 5.

Comments6: L76-82: Considering those presented in the introduction and the hypothesis/objectives of presented research manuscript, in my opinion, there are weak connections established between NO3- signaling and laccase activation; also, the manuscript should better justify the selection of CgLAC3

Response6: Thank you for your question and suggestion. We demonstrated that CgLAC3 promoter can respond to NO3- signal through tobacco transformation technique in the Figure5.

Comments7: L257: In the methods section, the manuscript presents the studied plant materials: Pomelo fruits (Citrus grandis cv Sanhongyou)–L 258; Pear (Pyrus spp. cv. Dangshansuli)–L268; Uniform soybean (Glycine max. cv. Huachun NO.3)-L270; Tobacco (Nicotiana benthamiana)–L274; In my opinion, in the manuscript should justify and motivate:Justification: Why the authors has selected these specific heterologous systems; Motivation: The significant phylogenetic distance between pomelo and the other species didnt compromise direct functional comparisons; How the different ways of tissue organisation and developmental patterns affect lignification processes

Response7: Thank you for the comments. Here, we selected pear as our primary research subject because that pear can form abundant of stone cells, which could be used as an excellent model for studying lignification characteristics. Additionally, as a non-fruit-bearing plant species, soybean roots also could serve as a valuable material for lignification research.

Comments8: L267-270: It is not clear how pears were treated with NO3- (immersions, spraying?). The authors should present clear details for: Please justify the selection of these concentration (0.01 and 0.15 g NO3-; L268); Why soybean was exposed to other concentration (0.2 and 20  g NO3-; L 271); Details on growth conditions are missing (light intensity, photoperiod, humidity, temperature, etc); What standardization metrics were considered for plant developmental stages?; When case, please provide details about soil composition and sterilization procedures;bFor laboratory experiments what acclimation period was considered before treatments? Also please detail how was verified (if it was verified) the NO3- uptake rate; and if pH adjustment was considered or not for the nutrient solution;

Response8: Thanks for your questions. Pears were treated with exogenous NO3- in the Supplemental Data Figure 3. The experiment of soybean was conducted in hydroponics, with air relative humidity of 60%, temperature of light 26℃/ dark 24℃, photoperiod of light 14 h/ dark 8 h. We determined the concentrations of NO3- and lignin in the pear in Supplemental Data Figure 4. For soybean roots, the pH value was used in the manuscript by Li et al., (PLoS Biology, 2022, 20: e3001739) entitled “Shoot-to-root translocated GmNN1/FT2a triggers nodulation and regulates soybean nitrogen nutrition”.

Comments9: L300: In my opinion, different tissue types used across species (roots vs. fruits) complicate comparative analyses; the authors should present how they overcome these challenges both in applied analytical methods and in results relevance and interpretation/comparisons. Considering the applied method, what kind of quality control assurance was performed considering the varied tissues/species?

Response9: Thank you for your question and suggestion. We agree with the experts that comparative analysis of tissue types in different species is complicated. Considering the phenotypic differences of plants, we studied the phenotypic differences of the same part through a single variable.

Comments10: The authors should be more specific; they should also provide details about control and the number of replicates considered

Response10: Thank you for your question and suggestion. We marked the quantities below the figures in the article (e.g. Figure1 A-H: n =6,)

Comments11: L 83: Considering the Results section in my opinion: Control groups are not well presented and compared. This diminishes the assessment of treatment-specific effects relevance; Time-dependent changes are not clearly presented; Overall, the results lack detailed comparisons between different tissues or conditions that could provide further mechanistic insights

Response11: Thank you so much for your thoughtful review on our research. We truly appreciate your attention to details. From field to lab, we demonstrated the positive correlation existed between nitrate and lignin concentration of pomelo juice sacs. We compare differences within the same species or organization using a single variable, and describe and measure them (Line 265-267, Line 270-273, Line 275-276).

Comments12: For the discussion section (L198) in my opinion, the authors should: more adequately and accurately integrate and compare their results with recent literature; more better consider alternative biochemical pathways that influence lignification; proposed mechanistic insights, specifically regarding NO3-and CgLAC3, shoul provide more detailed molecular explanations; the extrapolation from gene expression to functional outcomes should more adequately justified and provide supporting evidence; better consider the time-dependent aspects of lignification responses, in my opinion this is crucial for dynamic process interpretation

Response12: Thank you for your question and suggestion. We had compared lignin concentration of the rape or apple under different N level (Line 205-212). And lignification of pomelo fruits were showed in time-dependent aspects of lignification responses.

Dynamic changes of lignin concentration in fruit juice sacs

Comments13: L340-344: The conclusion section should be more clear and accurately presented; in my opinion, in the current version of the manuscript, this is poorly treated by the authors

Response13: Thank you for your question and suggestion. We have added to the conclusions in line336-341.

Reviewer 2 Report

Comments and Suggestions for Authors

There are some comments that might be helpful to improve the manuscript. This ms need major revise before accept.

1. It is recommended to supplement the statement on the correlation between lignin biosynthesis-related genes and nitrogen utilization in the Introduction section.

2.  Add  RNA-seq data in Results can enhances research quality.

3. The writing style of some genes requires italicization.

4. Some of the references are outdated.

5. It is recommended to add a diagram to help readers understand better.

6.  In the whole discussion, limited new idea or no deep thoughts.

Author Response

Dear Editors and Reviewers:

Thank you very much for the valuable comments on our manuscript entitled “Nitrate modulates fruit lignification by regulating CgLAC3 expression in pomelo” (Manuscript ID: 3595479). Those comments are all valuable and very helpful for revising and improving our paper, as well as the important guiding significance to our researches. We have studied comments carefully and have made revisions under revision mode in the resubmitted manuscript which we hope meet with approval. The corrections in answer: to the reviewer’s comments are listed one by one as follows:

Reviewer 2 

Comments1: It is recommended to supplement the statement on the correlation between lignin biosynthesis-related genes and nitrogen utilization in the Introduction section.

Response1: Thank you for your question and suggestion. We had added the lignin biosynthesis-related genes and nitrogen utilization in the discussion section (Line 205-212). (e.g.Wang et al. Multi-omics analysis reveals the mechanism of bHLH130 responding to low-nitrogen stress of apple rootstock. Plant Physiology 2023)

Comments2: Add RNA-seq data in Results can enhances research quality.

Response2: Thank you for your question and suggestion. We had added RNA-seq data in the Supplementary Data Table2.

Comments3: The writing style of some genes requires italicization.

Response3: Thank you for your question and suggestion. We have marked the genes in italics and the proteins in bold in the article.

Comments4: Some of the references are outdated.

Response4: Thank you for your question and suggestion. We checked and updated some references in the article. We also marked some content which are the following references with red color in the article (Line 44-48, Line 210-212, Line 244-246, Line 269-270).

e.g.

  • Miao,et al. Transcriptome analysis of nitrate enhanced tobacco resistance to aphid infestation. Plant physiology and biochemistry : PPB, 2025;
  • Huanget al. A comprehensive review of segment drying (vesicle granulation and collapse) in citrus fruit: Current state and future directions. Scientia Horticulturae 2023;
  • Liet al. LACCASE35 enhances lignification and resistance against Pseudomonas syringae actinidiae infection in kiwifruit. Plant physiology, 2025;
  • Menget al. PbMADS49 Regulates Lignification During Stone Cell Development in 'Dangshansuli' (Pyrus bretschneideri) Fruit. Plant, Cell and Environment, 2025;

Comments5: It is recommended to add a diagram to help readers understand better.

Response5: Thank you for your advice. We added a proposed model in the Figure 7.

Comments6: In the whole discussion, limited new idea or no deep thoughts.

Response6: Thank you for your question. In the whole discussion, we focus on that NO3--N might cause severe lignification and thus reduce fruit quality, and we suggest to avoid applying excessive amounts of N in practice so as to reduce over accumulation of NO3--N of fertilizer in order to minimize lignification and thus improve fruit quality (Line 225-226).

Round 2

Reviewer 1 Report

Comments and Suggestions for Authors

Dear Authors,

Please find my recommendation for the manuscript “Nitrate modulates fruit lignification by regulating CgLAC3 expression in pomelo”

  • Please justify the selection of fruit and plant species for study
  • The discussions section should be extended. Comparisons of the obtained results with other related studies should be performed and presented in the manuscript before consideration for potential publication in IJMS

Author Response

Dear Editors and Reviewers:

Thank you very much for the valuable comments on our manuscript entitled “Nitrate modulates fruit lignification by regulating CgLAC3 expression in pomelo” (Manuscript ID: 3595479). We have studied comments carefully and have made revisions under revision mode in the resubmitted manuscript which we hope meet with approval. The corrections in answer: to the reviewer’s comments are listed one by one as follows:

Reviewer 1 

Comments 1: Please justify the selection of fruit and plant species for study.

Response 1: Thank you very much for your suggestion. Given the whole plant transformation of pomelo is still under establishment, we choose pear fruit and soybean hairy roots for functional analysis with several reasons. On the one hand, pear is rich in stone cells, serves as an excellent model for studying lignification. Besides, the transient transformation with pear fruit has been widely employed to explore gene functions in fruit trees. On the other hand, to further reveal the function of CgLAC3 in regulating lignification in respond to nutrient, the soybean transgenic hairy roots as non-fruit plant species was used to solid the results. Our results demonstrated that CgLAC3 positively regulates lignin accumulation in planta.  

Moreover, to elucidate how NO3--N affects lignin formation, except for pomelo, we also choose the crop, pear and soybean, as plant material for further studies. We also want to see whether NO3--N triggered lignin accumulation is a common effect in both fruit and crop or not. The results demonstrate that high NO3--N promotion of lignification is common in plants (Line 119-121). Accordingly, we add some necessary descriptions in revised version. Please see Line 276-278 (pear), and Line 283-284 (tobacco). 

Comments 2: The discussions section should be extended. Comparisons of the obtained results with other related studies should be performed and presented in the manuscript before consideration for potential publication in IJMS

Response 2: Thank you for your question and suggestion. We presented a point of NO3--N may play pivotal roles in regulation of lignification in plants (Line 221-224) and added some LACs in the discussion section in the Line 245-252.

Reviewer 2 Report

Comments and Suggestions for Authors

The author has changed my questions and agreed to publish it

Author Response

Dear Reviewer,

Thank you so much for your kind feedback. I'm extremely grateful for your prompt and positive response, reviewer. Your approval means a lot to me, and I'm thrilled that the revised version meets your expectations for publication.

Sincerely yours,

Changhong Lai, Huiwen Zhou and Hong Liao

Root Biology Center, Fujian Agriculture and Forestry University, Fuzhou, 350002, China